# Quality of Life and Mood Status Disturbances in Cohabitants of Patients with Alopecia Areata: A Cross-Sectional Study in a Spanish Population

**DOI:** 10.3390/ijerph192316323

**Published:** 2022-12-06

**Authors:** Manuel Sánchez-Díaz, Pablo Díaz-Calvillo, Clara-Amanda Ureña-Paniego, Alejandro Molina-Leyva, Salvador Arias-Santiago

**Affiliations:** 1Dermatology Unit, Hospital Universitario Virgen de las Nieves, 18002 Granada, Spain; 2Instituto de Investigación Biosanitaria ibs.GRANADA, 18002 Granada, Spain; 3Trichology Clinic, Hospital Universitario Virgen de las Nieves, 18002 Granada, Spain; 4Dermatology Department, School of Medicine, University of Granada, 18016 Granada, Spain

**Keywords:** Alopecia Areata, cohabitants, quality of life, anxiety

## Abstract

A poor quality of life has been described in patients suffering from Alopecia Areata (AA). However, there is little evidence on how AA can impact on those living with patients. The aim of this study is to analyze the impact of AA on a cohabitant’s quality-of-life, mood status disturbances and sexual satisfaction. This is a cross-sectional study of AA patients and their cohabitants. Socio-demographic variables and disease severity, the quality of life, mood status disturbances and sexual dysfunction were collected using validated questionnaires. Eighty-four subjects were included in the study: 42 AA patients and 42 cohabitants. A poor quality of life and worse disease control in the patients were associated with a poorer quality of life of the family, higher scores of anxiety and depression, and the lower sexual satisfaction of cohabitants (*p* < 0.05). Anxiety and depression in patients were associated with worse family quality of life, higher rates of anxiety and less sexual satisfaction in cohabitants (*p* < 0.05). To conclude, AA seems to have an impact on the quality of life of cohabitants, leading to increased rates of anxiety, depression, a poorer quality of life, and reduced sexual satisfaction. In light of the results, a global approach for AA patients, including the care of the people who live with them, should be implemented.

## 1. Introduction

Alopecia Areata (AA) is an autoimmune, common and non-scarring type of alopecia with an unpredictable evolution [1]. Disease severity is variable, with patients having small well-defined alopecic patches on the scalp and others suffering from severe cases of total loss of all hair [2]. Treatment options are diverse, including topical treatments [1], oral corticosteroids [3], immunosuppressants [4] and, more recently, Janus kinase inhibitors [5]. However, the management of the disorder is still challenging, with a proportion of patients not achieving satisfactory responses to treatment. Moreover, due to the social relevance of hair, AA has previously been linked to a variety of impairments in the quality of life [6], mood status disturbances [7], and dysfunctional ways of communicating emotions [8].

Despite the impact of AA on patients’ quality of life having been addressed in previous studies [9], there is still a lack of studies addressing the potential burden that AA could pose on cohabitants. Exploratory studies have shown that other skin diseases, such as acne [10], psoriasis [11], or hidradenitis suppurativa [12], lead to a negative impact on those living with patients. In this regard, a worse quality of life, anxiety and depression, sexual and sleep disturbances, and altered personality traits, have been described.

Understanding which determinants are linked to a worse quality of life on those living with the patient, as well as those factors related to psychologic disturbances or dysfunctional personality traits, could be of great interest in order to implement a more global management of the disease. In this sense, not only the patient’s disease severity and quality of life, but also the impact these have on their family background, should be taken into account.

Therefore, this study aims to determine the presence of a poor quality of life, emotional state disturbances, and the loss of sexual function in cohabitants of patients with AA. Moreover, it aims to assess the potential associations between the quality of life of the patient and their cohabitant’s family quality of life, the cohabitant’s emotional state disturbance, the loss of sexual function, and personality traits. Finally, it aims to assess the potential role of the patient’s severity and the evolution time of the disease on the cohabitant’s quality of life, the cohabitant’s emotional state disturbances, the loss of sexual function, and personality traits.

## 2. Materials and Methods

Design: A cross-sectional study was performed. Patients diagnosed with AA, from mild to severe cases, and their cohabitants were included in order to assess the potential burden that AA could imply on the quality-of-life of cohabitants.

Patients: Subjects included in the study were recruited from the Trichology Clinic of the Virgen de las Nieves University Hospital. Both patients and cohabitants were asked to complete an online questionnaire after their protocolized follow-up consultation. Patients were recruited between September 2021–September 2022.

Inclusion criteria: The inclusion criteria for patients were: (a) patients with clinical diagnosis of AA, regardless of their treatment and disease severity; (b) patients aged 18 years old or older; and (c) patients must have given their informed consent.

The inclusion criteria for cohabitants were: (a) the cohabitants of patients with AA who were already enrolled in the study; (b) with independence of their marital status, the patient and the cohabitant had to be couple; (c) cohabitants had to be aged 18 years old or older; and (d) cohabitants must have given their informed consent.

Exclusion criteria: The exclusion criteria were: (a) refusal from the participants to be included in the study; and (b) participants suffering from major diseases apart from AA, which could affect their quality of life. The following disorders were considered: (i) active oncological diseases; (ii) any disorder which limited daily activities or generated a remarkable burden of disease, including neurologic, metabolic, digestive, musculoskeletal, cardiopulmonary or urinary diseases; (iii) mental disorders, which were diagnosed prior to the development of AA; and (iv) other skin diseases, different from AA, which led to an impairment on the quality of life.

Ethics: The present study was approved by the Research Ethics Committee of “Hospital Universitario Virgen de las Nieves” (internal code 1672-N-21) and is in accordance with the Declaration of Helsinki.

### 2.1. Variables of Interest

#### 2.1.1. Main Variables

Variables in relation to the severity of the disease and variables related to the quality of life were considered as main variables. Firstly, the collected variables in relation to disease severity were:The severity of Alopecia Tool II (SALT II): It was used to assess the disease severity by means of the percentage of the scalp affected by AA [13].Other disease characteristics such as the age of onset, disease duration, and treatments being used, were collected.

Secondly, variables in respect to different spheres of the quality of life, mood status disturbances, personality, sleep and sexual dysfunction, both in patients and cohabitants, were collected using validated questionnaires:Dermatology Life Quality Index (DLQ) was the collected skin quality of life marker. It includes 10 questions on a Likert scale from 0 to 3 each, with a final punctuation of 0 meaning no affectation, and 30 meaning the poorest quality of life [14].Family Dermatology Life Quality Index (FDLQI) was considered as the marker of cohabitant’s quality of life. It includes 10 questions on a Likert scale from 0 to 3 each, with a final punctuation of 0 meaning no affectation, and 30 meaning the poorest quality of life [15].DS14 Questionnaire was used to detect the presence of Type D Personality (TDp). There are 14 questions, 7 for negative affectivity and 7 for social inhibition, which are answered in a Likert scale. The cut-off point for considering TDp was a score ≥10 in the two spheres [16,17].Hospital Anxiety and Depression Scale (HADS) was used to assess the presence of anxiety and depression symptoms. It includes 14 items, which are scored in a adapted Likert scale, with 7 questions related to anxiety and 7 questions related to depression. A score ≥8 on any of the subscales was considered as the cut-off point for anxiety or depression, respectively [18].International Index of Erectile Function (IIEF-5) [19] and Female Sexual Function Index (FSFI-6) [20] questionnaires were employed to determine the presence of sexual dysfunction. IIEF-5 loss of sexual function in males, with ≤21 points being considered as an impairment on sexual function. The FSFI-6 evaluates female sexual impairment with scores ≤19 being considered as indicative of sexual dysfunction.Numeric Rating Scale (NRS) for sexual impairment was employed in a scale from 1 to 10 to detect the degree of sexual dysfunction associated with AA, as it has been previously reported [21].

#### 2.1.2. Other Variables

Demographic variables such as age, sex, marital status, and educational degree were collected. Additionally, previous diseases and the history of treatments used for A were recorded. Finally, questions about exclusion criteria were included in the survey so as to detect the presence of exclusion criteria.

### 2.2. Statistical Analysis

Descriptive statistics were used to analyze the sample characteristics. The Shapiro-Wilk test was applied to determine the normality of the data. Mean and standard deviation (SD) were used to report continuous data. Relative and absolute frequency distributions were used for qualitative data. The χ^2^ test or Fisher’s exact test were employed to make comparisons between nominal variables, and Student’s *t* test or the Wilcoxon-Mann-Whitney test were used to perform comparisons between nominal and continuous data. To further investigate potential related factors, associations between continuous data was evaluated by simple linear regression. The β coefficient and SD were considered as predictors of estimated effect of the independent variable on the dependent variable. p values less than 0.05 were considered for establishing statistical significance Statistical analyses were performed with JMP version 14.1.0 (SAS institute, Cary, NC, USA). The Bonferroni correction was applied for multiple comparisons.

## 3. Results

### 3.1. Sociodemographic and Clinical Characteristics of the Sample

Eighty-four subjects were included in the study: 42 AA patients and their corresponding 42 cohabitants.

#### 3.1.1. Patient’s Characteristics

The mean age of the patients was 40.59 (SD 8.92, range 18–73), and female to male ratio was 3.66. The majority of the patients were actively working and had a university or professional degree. Mean disease duration was 10.95 (SD 11.19), with 35.7% (15/42) of the patients having a prolonged disease (>10 years). Data regarding the quality of life survey can be seen in Table 1. Anxiety and depression were detected in 47.6% (20/42), and 47.6% (20/42) of the patients, respectively. Impaired sexual function was found in 69% of the patients (29/42): 78.8% (26/33) females and 33.3% (3/9) males.

#### 3.1.2. Cohabitant’s Characteristics

The mean age of the cohabitants was 46.76 (SD 12.10, range 18–68), and female to male ratio was 0.44. As was found for the patients, the majority of the cohabitants were actively working and had a university or professional degree.

Data regarding the quality of life questionnaires can be seen in Table 2. The rates of anxiety and depression in the cohabitants were 14.28% (6/42) and 11.9% (5/42), respectively, therefore being lower than the rates found for the patients (*p* < 0.01). Sexual dysfunction was detected in 57.1% of the cohabitants (24/42): 60% (9/15) females and 50% (15/30) males. The cohabitants family quality of life scores (FDLQI), anxiety and depression were found to be not associated to the cohabitant’s age, gender, educational level, nor occupation (*p* > 0.20).

### 3.2. Analysis of the Impact of Patient’s Quality-of-Life Indexes on Quality-of-Life Scores, Emotional Status and Sexual Function in Cohabitants

An exploratory analysis was performed to explore the potential correlation between the quality of life in patients (DLQI) and the quality of life (FDLQI), emotional status (HADS), sexual dysfunction (IIEF and FSFI), and type D personality (DS14) of cohabitants (Table 3). A poor quality of life in patients was found to be related to poor quality of life scores of their couples (FDLQI, *p* = 0.003) (Figure 1). Moreover, it was associated with higher anxiety scores (*p* = 0.03), and the lower sexual satisfaction of cohabitants (NRS for sexual impairment, *p* < 0.001). On the other hand, depression scores in the cohabitants cohabitant TDp, and cohabitant sleep quality, were found to be not related to patient quality of life.

### 3.3. Analysis of the Impact of Disease Severity on Quality-of-Life Scores, Emotional Status and Sexual Function in Cohabitants

This study explored the association between poorer disease control (SALT score) and indicators of the quality of life, altered mood, sexual impairment, and type D personality of the cohabitants (Table 3). Worse disease control was found to be associated with worse family quality of life (*p* = 0.03) (Figure 1), higher cohabitant anxiety scores (*p* = 0.002), and depression scores (*p* = 0.05). No associations were found for cohabitant depression, type D personality, sexual impairment, nor sleep impairment.

### 3.4. Analysis of the Impact of Disease Duration on Quality-of-Life Scores, Emotional Status and Sexual Function in Cohabitants

The relationship between disease duration and the quality of life scores, emotional status scores, TDp and sexual impairment in cohabitants was explored. (Table 3). A longer disease was found to be associated with higher cohabitants sexual impairment (*p* = 0.02). However, no associations were found for family quality of life, type D personality, cohabitant anxiety or depression, nor the specific subscales of sexual impairment.

### 3.5. Analysis of the Impact of Patient’s Emotional Status on Quality-of-Life Indexes, Emotional Status and Sexual Function in Cohabitants

The association between HADS-A and HADS-D scores and the quality of life indexes, mood disturbances, sexual disfunction, and type D personality of cohabitants, was explored (Table 3). Higher anxiety scores in patients with AA were associated with poorer family quality of life (*p* < 0.01) (Figure 1), higher anxiety scores in cohabitants (*p* = 0.02), higher sexual impairment in cohabitants (*p* = 0.002), and tending towards statistical significance, specific female sexual impairment scores (*p* = 0.08).

On the other hand, higher depression scores in patients with AA were associated with poorer family quality of life (*p* < 0.01) (Figure 1), higher anxiety and depression scores in cohabitants (*p* < 0.001), and higher sexual impairment scores in cohabitants (*p* = 0.002). No other significant associations were found.

## 4. Discussion

AA is a skin disease, which has been previously linked to a poor quality of life in patients [22,23]. However, to date, no studies have assessed the burden that this disease has on the patient’s cohabitants. The present study aimed to explore for the first time the burden that this disease could represent for the cohabitants of AA patients. In light of the results, it has been found that worse disease control, a poorer quality of life, and higher anxiety and depression rates in patients are associated with a poorer family quality of life, higher rates of anxiety and depression, and lower sexual satisfaction in cohabitants. Moreover, long-lasting disease seems to be related to worse sexual satisfaction in cohabitants.

Previous studies have addressed the issue of the quality of life of cohabitants of patients suffering from chronic diseases, such as inflammatory bowel disease [24], psoriasis [11], chronic spontaneous urticaria [25], or hidradenitis suppurativa [12]. In this line, there are many factors that could be responsible for this negative impact. First of all, poorer disease control could lead to a greater need for care in the home setting, as well as higher rates of frustration in patients and their couples. This may negatively impact the quality of life, as well as rates of anxiety and depression. Therefore, the present study corroborates the data already obtained for other pathologies.

On the other hand, the duration of the disease could play a role in the negative impact of the disease on the cohabitants of patients with AA. In this regard, we have observed that a longer duration of the disease is associated with a poorer assessment of the sexual life of the partners of patients with AA. The remitting and recurrent course of the disease could generate frustration in patients and their partners, thus diminishing their appreciation of their sexual life. Future studies specifically assessing which factors of longer disease duration influence the poorer quality of life of cohabitants would be of great interest. To our knowledge, there are no reports of similar associations for other diseases.

In the light of the results, the development of new effective treatments would be of great benefit for patients and cohabitants. A better control of the disease would probably improve the quality of life of the patients, which would decrease the impact of the disease on the cohabitants of the patients suffering from AA.

Finally, our study shows how anxiety and depression in patients is associated with a negative impact on cohabitants in terms of a poorer quality of life, and higher rates of anxiety and depression. To the best of our knowledge, already available scientific literature for other chronic diseases have not addressed this issue [12,24]. Social and psychological support for patients would probably improve their mood status disturbances, therefore resulting in a better quality of life for cohabitants, as well as lower rates of anxiety and depression.

The main limitations of the present study are: (a) the sample size, which could have limited the detection of significant differences; (b) the cross-sectional design, which makes it impossible to assess causality; (c) the inclusion only of the patients’ partners, without including other types of cohabitants; and (d) the lack of a control group

## 5. Conclusions

To conclude, AA seems to have an impact on the quality of life in cohabitants, especially in terms of the increased rates of anxiety, depression, a poorer quality of life, and reduced sexual satisfaction. Furthermore, longer disease times have been associated with lower sexual satisfaction in the patients’ partners. These data could help to propose a more global model of medical care for AA patients, that includes not only the medical and psychological alterations of the patients, but also those of the people who live with them.

## Figures and Tables

**Figure 1 ijerph-19-16323-f001:**
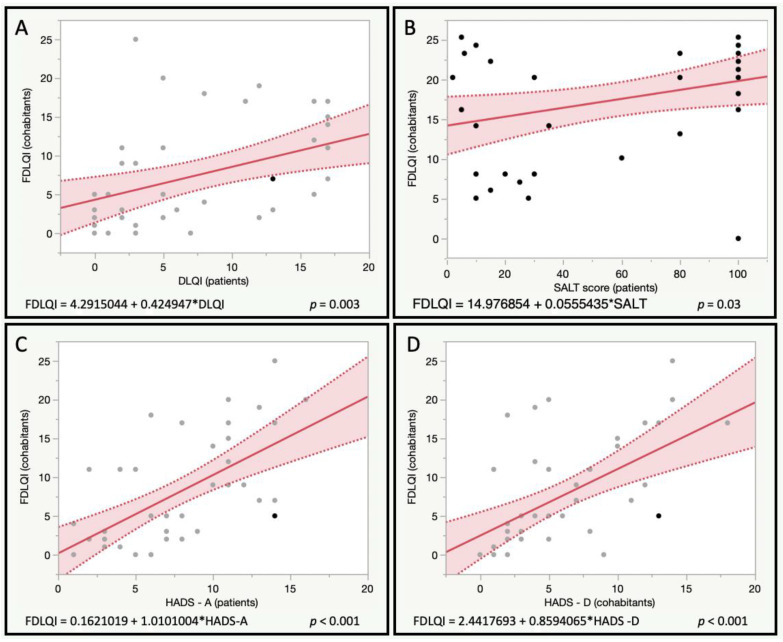
Correlation of Family Dermatology Life Quality Index (FDLQI) with patient’s quality of life (**A**) (Dermatoloqy Life Quality Index—DLQI-), severity of the disease (**B**) (Severity of Alopecia Tool—SALT), and patient’s scores for anxiety and depression (**C**,**D**) (Hospital Anxiety and Depression Scales—HADS).

**Table 1 ijerph-19-16323-t001:** Sociodemographic features of the patients, characteristics of the disease and quality of life indicators.

Variables Patients (N = 42)					
Socio-demographic features					
Age (years)	40.59 (SD 13.62)		Occupation	Employed	57.1% (24/42)
				Unemployed	42.9% (18/42)
Sex (%)	Male:	21.4% (9/42)	Educational level	No studies or compulsory education	28.6% (12/42)
	Female:	78.6% (33/42)		Professional or university studies	71.4% (30/42)
Disease characteristics					
Disease duration (years)	10.95 (SD 11.19)		SALT score	57.45 (SD 41.62)	
Disease duration	<10 years	64.3% (27/42)	Current treatment for AA	No treatment/topical/ intralesional treatments	71.4% (30/42)
	>10 years	35.7% (15/42)		Oral corticosteroids	16.7% (7/42)
				Immunosuppressive agents/JAK inhibitors	11.90% (5/42)
Quality of life indicators					
DLQI	8.16 (SD 7.19)		DS14 (% of positive test)	35.7% (15/42)	
HADS Depression (% of positive test)	73.8% (31/42)		HADS Anxiety (% of positive test)	47.6% (20/42)	
FSFI (% of female sexual dysfunction)	78.8% (26/33)		IIEF (% of male sexual dysfunction)	33.3% (3/9)	

**Table 2 ijerph-19-16323-t002:** Sociodemographic features of the cohabitant sample and quality of life indicators.

Variables Cohabitants (N = 42)					
Socio-demographic features					
Age (years)	46.76 (SD 12.10)		Occupation	Employed	69% (29/42)
				Unemployed	31% (13/42)
Sex (%)	Male:	69% (29/42)	Educational level	No studies or compulsory education	47.6% (20/42)
	Female:	31% (13/42)		Professional or university studies	52.4% (22/42)
Quality of life indicators					
FDLQI	7.76 (SD 6.87)		DS14 (% of positive test)	4.7% (2/42)	
HADS Depression (% of positive test)	11.9% (5/42)		HADS Anxiety (% of positive test)	14.28% (6/42)	
FSFI (% of female sexual dysfunction)	60% (9/15)		IIEF (% of male sexual dysfunction)	50% (15/30)	

**Table 3 ijerph-19-16323-t003:** Correlation between quality of life indexes of the cohabitants and patient’s DLQI and SALT.

Factors	Patient DLQI		Patient SALT Score		Patient Disease Duration		Patient Anxiety (HADS-A)		Patient Depression (HADS-D)	
	Mean/Beta	*p* Value	Mean/Beta	*p* Value	Mean/Beta	*p* Value	Mean/Beta	*p* Value	Mean/Beta	*p* Value
FDLQI	0.42 (SE 0.13)	0.003 *	0.05 (SE 0.02)	0.03	0.11 (SE 0.09)	0.21	1.01 (SE 0.19)	<0.001 *	0.85 (SE 0.19)	<0.01
Cohabitant Anxiety (HADS-A)	0.17 (SE 0.08)	0.03	0.04 (SE 0.01)	0.002 *	0.03 (SE 0.05)	0.51	0.32 (SE 0.13)	0.02	0.42 (SE 0.11)	<0.001 *
Cohabitant Depression (HADS-D)	0.09 (SE 0.08)	0.24	0.04 (SE 0.01)	0.05	0.02 (SE 0.05)	0.66	0.19 (SE 0.13)	0.14	0.41 (SE 0.11)	<0.001 *
Cohabitant DS14 score	−0.24 (SE 0.24)	0.31	−0.06 (SE 0.04)	0.14	0.09 (SE 0.15)	0.53	−0.29 (SE 0.40)	0.47	−0.05 (SE 0.39)	0.89
Cohabitant NRS for sexual impairment	0.21 (SD 0.06)	0.001 *	0.01 (SD 0.01)	0.37	0.09 (SE 0.04)	0.02	0.33 (SE 0.10)	0.002	0.37 (SE 0.09)	0.002
Male cohabitant Sexual impairment index (Men—IIEF)	−0.01 (SD 0.10)	0.88	0.02 (SD 0.01)	0.21	0.02 (SE 0.07)	0.70	−0.24 (SE 0.l9)	0.20	−0.18 (SE 0.16)	0.27
Female cohabitant sexual impairment index (Women—FSFI)	−0.27 (SD 0.64)	0.68	−0.04 (SD 0.07)	0.61	−0.25 (SE 0.25)	0.35	−1.13 (SE 0.57)	0.08	−0.92 (SE 0.67)	0.20

DLQI—Dermatology Quality of Life Index; FDLQI: Family Dermatology Quality of Life Index; FSFI: Female Sexual Function Index; HADS: Hospital Anxiety and Depression Scale (A—Anxiety; D—Depression); IIEF: International Index of Erectile Function; SALT: Severity of Alopecia Tool. *—*p* values within limits of significance after applying the Bonferroni’s modification for multiple comparisons.

## Data Availability

Data are available upon reasonable request.

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
