# Peer review of "Quality of Life and Mood Status Disturbances in Cohabitants of Patients with Alopecia Areata: A Cross-Sectional Study in a Spanish Population"

_ijerph, 2022, doi:10.3390/ijerph192316323_

Round 1
Reviewer 1 Report
It seems a good piece of work. However i have a few suggestions that need to be addressed before publication.
The sample size is too small to project the conclusion for a large population. If this study is done on a specific region or part of the state that need to be highlighted in the title.
Author Response
Estimated reviewers,
I would like to thank you for your comments, as they allow us to improve the scientific quality of our study. All the suggested changes have been performed. Below, you can see a point-by-point response to each suggestion.
Reviewer 1
Response: Thank you very much, the title has been modified, and the limitation of sample size has been included in the limitations.
Reviewer 2 Report
This is actually an interesting idea to determine the presence of poor quality of life, emotional state disturbances and the loss of sexual function in cohabitants of patients with alopecia areata. The article is quite interesting, although the results were obtained from a sample size number of cases. However, some major points should be addressed in this study.
1. Which is the range of age, from 18 to?
2. Regarding patients with the disease for more than 10 years, had they performed tests? (E.g. urological, gynaecological, psychiatric or psychological eamination)
3. In addition to the alopecia areata medications, were the patients taking any other medications?
4. Which is the control group?
5. There are few grammar mistakes. Check the legend of Table 3
Author Response
Estimated reviewers,
I would like to thank you for your comments, as they allow us to improve the scientific quality of our study. All the suggested changes have been performed. Below, you can see a point-by-point response to each suggestion.
Reviewer 2
1. Age range has been included in the results section.
2. Patients with disease >10 years, and also those <10 years are studied with protocoled tests, which include blood tests. Only those with symptomatology are referred to other specialists,.
3. Patients could be taking other medications to be included in the study unless they imply that they have a major disease which could significantly impact on their quality of life. Therefore, medications are not one of the exclusion criteria themselves, and so they were not systematically recorded in the study.
4. There is no "control group" in this study, as the idea is to correlate the activity indexes and quality-of-life scores in patients with quality-of-life and mood status disturbances in cohabitants. The lack of control group is therefore a limitation to generalize some results. This has been included in the discussion section to make the text more clear.
5. Grammar has been improved. Table 3 legend has been corrected, as it was wrongly written.
Reviewer 3 Report
Thank you so much for the opportunity to review this article aimed to analyze the impact of AA in patient and cohabitants quality of life indicators
The study design is adequate, the conclusions are well supported by results, the manuscript is clear and concise, and discussion is interesting. The main drawbacks I noted during the reading are all appointed by the authors as weak points.
I can only recommend avoiding some of the repeating data, well reported in tables, in the main text.
In summary, this study is interesting, well-supported, and carefully and clearly presented.
Author Response
Dear Reviewer,
We would like to thank you for your comments as they allow us to improve the scientific quality of our work. We have deleted some repeated information from the results section to make the text more easily readable.
Round 2
Reviewer 2 Report
The authors have addressed most of the reviewer's comments adequately.